# Squeezing Turbulence Statistics out of a Pulsed Doppler Lidar

Mohammadreza Manami[1,2], Jakob Mann[1], Mikael Sjöholm[1], Guillaume Léa[2], and Guillaume Gorju[2]

[1]DTU Wind and Energy Systems, Technical University of Denmark, Roskilde, Denmark
[2]Lidar Division, Lumibird SA, Lannion, France

**Correspondence:** Mohammadreza Manami (manami@dtu.dk)

**Abstract.** Accurate estimation of second-order turbulence statistics using pulsed Doppler lidar has been a challenge for a long time, mainly due to the negative influence of probe volume averaging. The present study aims to investigate a novel approach to extracting first- and second-order turbulence statistics directly from the average Doppler spectra in the frequency domain. The main hypothesis is that averaging Doppler spectra over 10-minute intervals can mitigate the influence of probe volume averaging and random noise in velocity retrievals, thereby improving estimates of velocity variance. To achieve this, we develop a new analytical model for the time-averaged Doppler spectrum, beginning with a theoretical formulation based on the beat signal within the range gate. The model is applied to 10-minute averaged Doppler spectra collected by a pulsed lidar system pointing toward a sonic anemometer mounted on a meteorological mast in front of a Vestas V52 wind turbine at the DTU Risø campus in Denmark. Validation results demonstrate that the Doppler spectra model, when fitted to 400 ns nominal pulse durations, closely matches sonic anemometer measurements in both mean radial velocities and standard deviations. This agreement is quantified by the least orthogonal square fit slopes of 0.976 for the mean velocities and 0.983 for the standard deviations. In comparison to the conventional time-domain approach, which accounts for only 72.1% of the standard deviation, the proposed spectral method captures 98.3% of the standard deviation observed in the sonic anemometer. However, this model does not accurately estimate variances using the short pulse (200 ns) of the instrument. Despite this limitation for the short pulse, the proposed method is an important step towards better turbulence estimation from pulsed Doppler lidars.

## 1 Introduction

Turbulence, characterized by the standard deviation of wind speed fluctuations, is crucial in determining the mechanical loads on wind turbines (Brand et al., 2011; Van Kuik et al., 2016; Veers et al., 2019). However, accurately measuring turbulence with pulsed lidar remains challenging due to factors such as probe volume averaging, cross-contamination, and high-frequency noise in the velocity estimation process, which degrade the accuracy of velocity variance measurements (Mann et al., 2008; Sathe and Mann, 2013; Sathe et al., 2015; Puccioni and Iungo, 2021).

The conventional method for measuring turbulence with pulsed lidar involves deriving velocity time series and computing variances relative to the mean velocity, but this approach presents the above challenges (Sathe et al., 2011). As an illustration of some of the challenges associated with lidar-based turbulence measurements, figure 1 presents an example highlighting the contrasting biases due to high-frequency noise and probe volume averaging on velocity variance estimates. The figure shows pre-multiplied turbulence spectra of radial velocity for a pulsed lidar, where different averaging levels have been applied

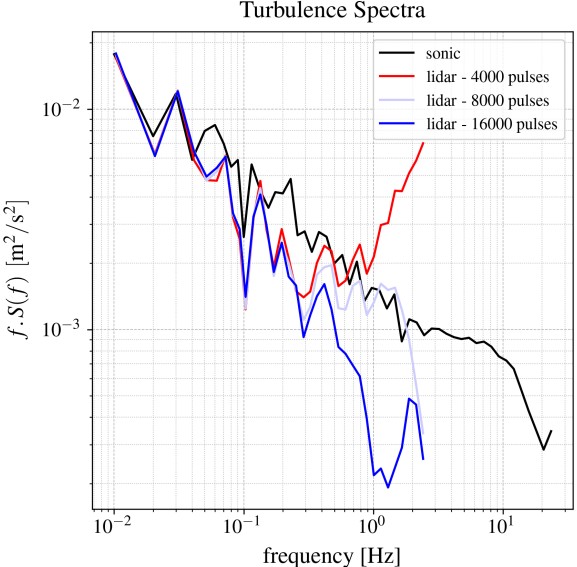

**Figure 1.** Turbulence spectra derived from the 10-minute radial velocity time series using pulsed lidar without additional averaging (red curve) and a sonic anemometer (black curve). The light and dark blue curves illustrate turbulence spectra obtained from moving averages of the Doppler spectra calculated over two and four times the original number of pulses.

to the Doppler spectra. The turbulence spectra from pulsed lidar are compared with a high-resolution turbulence spectrum derived from a collocated sonic anemometer at the corresponding lidar range gate. The original Doppler spectra (without additional averaging) are averaged over 4000 pulses (red curve), with a sampling frequency of 5 Hz. An additional averaging is applied through moving averages over two (light blue curve) and four times (dark blue curve) the original number of pulses. The reference spectrum from the sonic anemometer (black curve) is based on the projection of three-dimensional wind velocity, with a sampling rate of 50 Hz, onto the lidar beam direction. The comparison indicates that the spectra from the lidar and the sonic anemometer align up to a frequency of 0.03 Hz. However, as the number of averaged Doppler spectra increases, the lidar measurements exhibit reduced variance compared to those from the sonic anemometer, particularly at the high-frequency end of the original turbulence spectrum derived from the lidar. While the total variance observed in the experimental lidar measurements (without additional averaging) coincidentally aligns with the variance measured by the sonic anemometer, this appears due to erroneous reasons. The observed compensation in total variance results from the combined effect of increased variance caused by high-frequency noise and the reduced variance due to probe volume averaging. This arbitrariness underscores the inherent difficulty in obtaining accurate, noise-free variance estimates, which is the main objective of this study.

In this study, we explore an alternative technique for extracting turbulence information from pulsed lidar, inspired by a methodology previously applied to continuous-wave lidars (Branlard et al., 2013). While it has long been recognized that the width of the Doppler spectrum is partially linked to turbulence (Zrnic, 1979; Frehlich, 2013; Banakh and Smalikho, 2013), we

aim to establish the theoretical relationship. Our key hypothesis is that time-averaged Doppler spectra can yield more accurate

velocity variance estimates by reducing the impact of volume (or spatial) averaging as well as decreasing the impact of random noise in the measurement of the velocity fluctuations compared to the standard way of calculating statistics from a time series. The proposed method provides more robust estimates that could contribute to more cost-effective mechanical load assessments of wind turbines.

## 2   Methodology

To understand the properties of the Doppler spectrum of a pulsed lidar, we begin by analyzing the basic geometry of the detection process. The pulsed lidar transmits pulses in rapid succession, and for this analysis, we assume that these pulses have a Gaussian temporal profile with a full width at half maximum, $FWHM$, determined by the Gaussian fit. The Gaussian shape is a reasonable assumption for the Lumibird pulsed lidar under study. This assumption is confirmed by the measured pulse shapes after the photodiode for nominal pulse durations of 200 ns, 400 ns, and 800 ns, as depicted in figure 2. We assume

that the range gate $T$ is centered around $t = 0$, with the zero of the line-of-sight (LOS) axis set to align with the center of the corresponding range gate. The range gate is assumed to be rectangular, without any additional weighting. As illustrated in figure 3, only a scattering particle $p$ will scatter light into the range gate. Due to the LOS velocity $u$ of the particle, the scattered light will experience an angular frequency shift of $\Delta\omega = 2u\omega_0/c$, where $\omega_0$ is the emitted angular frequency and $c$ is the speed of light. We assume the scattered light has a complex amplitude $A$, which depends on the particle's properties. The

lidar typically averages thousands of spectra to compute a Doppler spectrum, usually over about one second. In this case, we focus on longer averaging periods, such as 10 minutes, to estimate the turbulence averaged over that time.

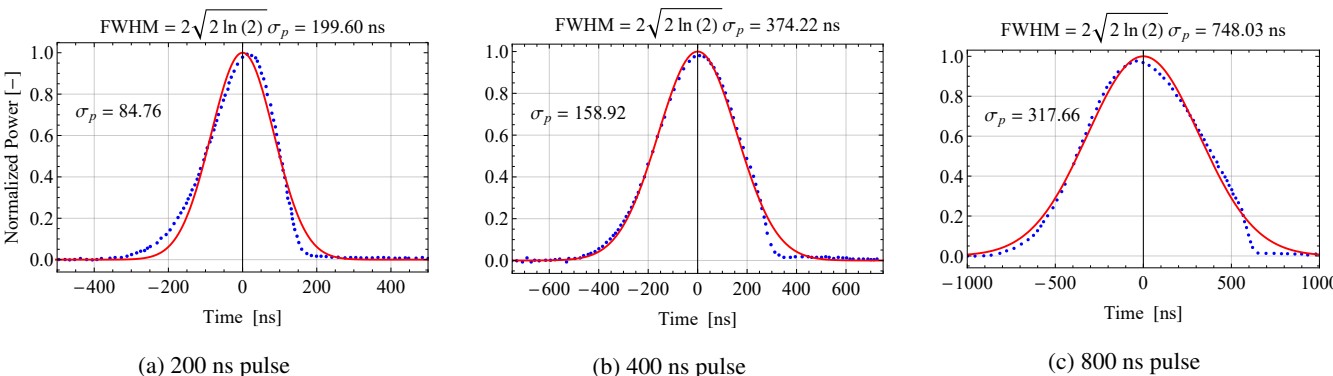

**Figure 2.** Measured laser power over time (blue points) with Gaussian fits (red curves) approximating the pulse shapes. Each pulse is normalized to its peak power.

We assume that the beat signal $a(t)$ in the range gate can be written in terms of a Gaussian pulse shifted by a frequency of $\Delta\omega$:

$$a(t) = \frac{A}{\sqrt{2\pi}\sigma_p} \exp\left(-\frac{1}{2}\frac{(t - \frac{2s}{c})^2}{\sigma_p^2}\right) \exp(i\Delta\omega t), \tag{1}$$

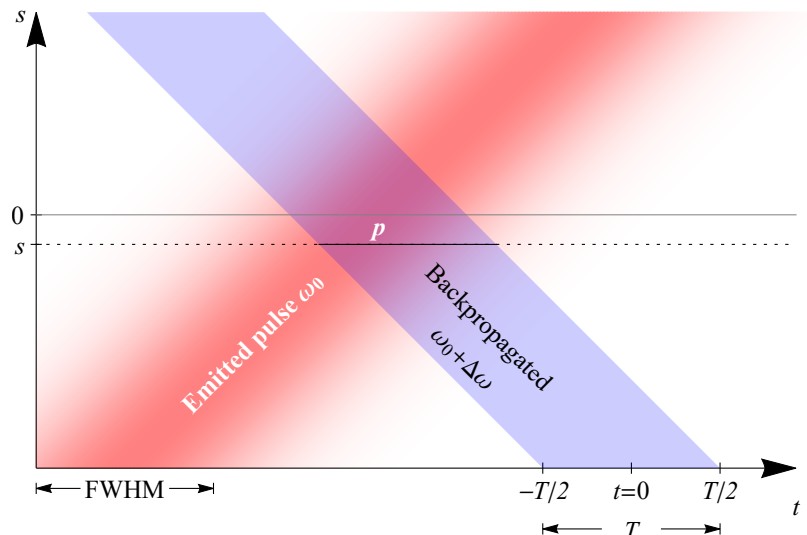

**Figure 3.** Time-space schematic of an emitted Gaussian pulse of duration $FWHM$ and angular frequency $\omega_0$. A particle $p$ is positioned at a LOS distance $s$. The back-scattered light from the particle with frequency shift $\Delta\omega$ is received in a range gate with duration $T$.

where a scattering particle is at a position $s$ and the Gaussian pulse has an amplitude $A$ and a standard deviation $\sigma_p$. If the particle is further away where $s = 0$ (see figure 3), then the maximum of the scattered pulse signal will arrive $\frac{2s}{c}$ later to the detector. We also assumed that the particle at position $s$ does not change significantly during exposure to the pulse, and the particle has a constant velocity. The frequencies at which the spectrum is obtained are given by:

$$\omega_l = l\frac{2\pi}{T} , \quad l \in \mathbb{Z} \tag{2}$$

Now, the average spectrum can be expressed as the ensemble average of the absolute squared Fourier transform of the beat signal:

$$S(\omega_l) = \left\langle \left| \int_{-T/2}^{T/2} \frac{A}{\sqrt{2\pi}\sigma_p} \exp\left( -\frac{1}{2}\frac{(t-\frac{2s}{c})^2}{\sigma_p^2} \right) \exp(i(\Delta\omega - \omega_l)t)dt \right|^2 \right\rangle \tag{3}$$

The ensemble average in the expression (3) for the average Doppler spectrum is over all particle positions $s$, all velocities $u$, and all amplitudes $A$. The particles appear randomly and uniformly along the range, and the $u$, $s$, and $A$ are assumed
independent. The distribution of velocities is assumed Gaussian with mean $\mu$ and standard deviation $\sigma$, and it does not change along the beam. The ensemble average representation in the spectrum can be expressed as an integral of probability densities (Wolf and Mandel, 1995):

$$S(\omega_l) = \int_0^\infty \iint_{-\infty}^\infty |\cdot|^2 p(s)p(u)p(A)dsdudA, \tag{4}$$

where $p(s)$, $p(u)$ and $p(A)$ are the probability density distributions of those variables. The probability density function of $s$, is considered to be uniform, and the probability density of $u$ is considered to be normal and independent of $s$:

$$p(u) = \frac{1}{\sqrt{2\pi}\sigma} \exp\left(-\frac{1}{2}\frac{(u-\mu)^2}{\sigma^2}\right), \tag{5}$$

where $\mu$ is the mean LOS velocity and $\sigma$ is the standard deviation.

The absolute square in equation (3) can be rewritten as a two-dimensional integral

$$\left|\int_{-T/2}^{T/2} f(t)dt\right|^2 = \iint_{-T/2}^{T/2} f(t)f^*(t')dtdt', \tag{6}$$

where $^*$ denotes complex conjugation. Thus, according to equation (4), $S(\omega_l)$ can be expressed as a five-fold integral. However, since amplitude is independent of all the other variables, the integral over $A$ can be pulled outside the five-fold integral and it is discarded because we are only interested in the shape of the spectrum, not its amplitude. Discarding all constant terms gives:

$$S(\omega_l) = \iint_{-\infty}^{\infty} \iint_{-T/2}^{T/2} \exp\left(-\frac{1}{2}\frac{(t-\frac{2s}{c})^2}{\sigma_p^2}\right) \exp\left(-\frac{1}{2}\frac{(t'-\frac{2s}{c})^2}{\sigma_p^2}\right) e^{i(\Delta\omega-\omega_l)(t-t')}p(u)p(s)dtdt'dsdu. \tag{7}$$

The integral over $s$ is a convolution of two Gaussians, which is again a Gaussian such that (discarding any constants):

$$S(\omega_l) = \int_{-\infty}^{\infty} \iint_{-T/2}^{T/2} \exp\left(-\frac{(t-t')^2}{4\sigma_p^2}\right) e^{i(\Delta\omega-\omega_l)(t-t')}p(u)dtdt'du. \tag{8}$$

We shall now perform the integral over $u$. Apart from $p(u)$, the only term in the integrand that depends on $u$ is $\Delta\omega = 2\omega_0 u/c$, which means that the integral is essentially a Fourier transform of a non-centered Gaussian and the result is:

$$S(\omega_l) = \iint_{-T/2}^{T/2} \exp\left(-D(t-t')^2\right) \exp(iB(t-t'))dtdt', \tag{9}$$

where

$$D = \frac{1}{4\sigma_p^2} + 2\omega_0^2\frac{\sigma^2}{c^2} = \frac{1}{4\sigma_p^2} + \frac{8\pi^2}{\lambda_0^2}\sigma^2, \tag{10}$$

$$B = 2\frac{\mu}{c}\omega_0 - \omega_l = \frac{4\pi\mu}{\lambda_0} - \frac{2\pi}{T}l \ . \tag{11}$$

This double integral can be reduced to a single integral by the following change of variables: $\tau = t - t'$ and $s = t + t'$, i.e., the so-called diamond transformation, and the final result becomes

$$S(\omega_l) \propto \int_0^T (T-\tau)e^{-D\tau^2}\cos(B\tau)d\tau. \tag{12}$$

The computationally efficient version of the above equation is as follows:

$$S(\omega_l) = \frac{-1 + e^{-\alpha}\cos\beta}{2\alpha} + \frac{1}{2\alpha^{3/2}}\left(\beta\,\mathrm{F}\left(\frac{\beta}{2\sqrt{\alpha}}\right) - \mathrm{Re}\left[(2i\alpha+\beta)e^{-\alpha+i\beta}\,\mathrm{F}\left(i\sqrt{\alpha}+\frac{\beta}{2\sqrt{\alpha}}\right)\right]\right), \tag{13}$$

where $F$ is the Dawson integral (Abramowitz and Stegun, 1972)

$$F(x) = e^{-z^2} \int_0^z e^{z^2} dz \,, \tag{14}$$

and $\alpha = DT^2$ and $\beta = BT$.

Equation (13) represents the simplified form of the averaged Doppler spectra model used for estimating turbulence statistics. The proposed model allows for the quantification of turbulence statistics across all scales. A significant advantage of this approach is its ability to capture velocity variances at scales smaller than the probe volume, a feature generally absent in the conventional method. The model considered the homogeneity assumption along the LOS, which neglects the shear effects. However, it does not rely on any isotropy assumptions that restrict the measurements to preferred directions.

This theoretical model is validated through field experiments using a pulsed lidar pointing toward a sonic anemometer. In the analysis, Doppler spectra obtained from the pulsed lidar are averaged over 10-minute intervals for two different pulse durations. The proposed model, equation (13), is then fitted to the averaged Doppler spectra to estimate the mean wind velocities and standard deviations. The downhill simplex method (Nelder and Mead, 1965; Nocedal and Wright, 2006) is chosen to minimize the objective function, which is defined as the least squares difference between the model's prediction and the peak section of the averaged Doppler spectra. The downhill simplex optimization method is a direct search algorithm that does not require analytical or numerical derivatives. However, this method is relatively slow in terms of computational efficiency. The estimated mean wind radial velocities and standard deviations from this model, equation (13), are validated with those obtained by projecting the sonic velocity onto the direction of the lidar beam. The following section provides a detailed overview of the field experiment.

## 3   Field Experiment

During the field experiment, Doppler spectra were collected using a prototype pulsed lidar system developed by Lumibird. This pulsed lidar has a telescope aperture diameter of 50 mm and operates at a wavelength of 1548 nm, with a pulse energy of 50 $\mu$J and a sampling rate of 250 MHz. This system was deployed near a meteorological mast used for measuring the westerly inflow to a Vestas V52 wind turbine at the Risø campus in Denmark (figure 4). The optimal wind direction according to the geometry of the experiment is the westerly sector, where the flow comes. The westerly flow comes from the fjord and remains unaffected by wake effects from either the V52 wind turbine or the met mast. The ground-based pulsed lidar was aligned using a telescopic sight to point at the sonic anemometers mounted on the north side of the V52 mast, as demonstrated in figure 5. To ensure accurate velocity projection, the coordinates of the lidar, boom, and sonic anemometers were recorded within the same coordinate system using the Leica Total Station (Leica Geosystems AG, 2013), allowing for the projection of three-dimensional sonic velocity measurements onto the lidar beam direction. The Leica Total Station can accurately measure distances with reflectors to an order of magnitude $O(10^{-4})$ meters within ranges of 1800 meters. After taking the distance and angular measurements, the device performs trigonometric conversions to transform the relative polar coordinates to the station

setup into cartesian coordinates. The local coordinates of the desired points will then be converted into the global reference system. The reference coordinate system was set to WGS 84 / UTM zone 32N for all recorded coordinates in our experiments.

This measurement campaign was carried out over several months, and various tests were performed. However, the period from October 17 to October 31, 2024, was selected for analysis of Doppler spectra due to the higher data availability. During this period, the lidar was pointing at the sonic anemometer positioned at a height of 18 meters. The radial distance from the lidar telescope to the base of the sonic anemometer is 117.45 m. Therefore, the gate centered at 120 m was selected for the Doppler spectra analysis, as it is the nearest gate to the sonic anemometer. Two different nominal pulse durations were considered in the

pulsed lidar: short (200 ns) and medium (400 ns), alternating every 20 minutes in this experiment. The acquisition time was set to 400 ms, with a pulse repetition rate of 20 kHz for short and 10 kHz for medium pulses.

## 4   Results and Analysis

As stated earlier, the experiment aimed to compare turbulence statistics obtained from fitting the model, equation (13), to the 10-minute averaged Doppler spectra with those measured by the sonic anemometer. Hereafter, the model fit to the averaged

Doppler spectra measured will be referred to as the model. Figure 6 illustrates the mean radial wind velocities and standard deviation for the two analyzed pulse lengths. The results demonstrate that the model's estimated mean wind velocities closely align with reference measurements from the sonic anemometer for both short and medium pulses. For the standard deviation, the model's estimations based on Doppler spectra of medium pulses generally follow the trend observed in the sonic data. However, when the standard deviation is low, the model occasionally estimates zero. This issue is more pronounced when

using Doppler spectra from short pulses, where zero values frequently appear at low standard deviations. Moreover, for short pulses, the model fails to capture the standard deviation trend observed in the sonic instrument. In particular, under laminar flow conditions, estimating velocity variances is expected to be challenging, since in this regime the primary contribution to the Doppler spectral width arises from the finite length of the range gate. However, obtaining zero variances from the model is clearly unrealistic. This outcome reflects the imperfection of the model, which relies on several simplifying assumptions.

One key assumption is that the pulse shape follows a Gaussian distribution, whereas, in practice, measured pulses often deviate from a Gaussian form. Another source of error arises from the background noise removal: when the noise level is subtracted, it can sometimes yield negative values in the Doppler spectra. The negative Doppler spectra are non-physical and introduce inaccuracies in the variance estimation.

In order to quantify the correlation between the model's estimations and the sonic measurements, the least orthogonal

squares fit was applied with a constraint forcing the fit through the origin. Since measurement errors may occur under easterly and southerly winds due to wake effects from the wind turbine and the met mast, data from the 90°–110° and 170°–200° sectors were excluded to ensure the validity of the comparison between the lidar and the sonic anemometer. The mean wind velocities in both cases strongly correlate with the sonic measurements, as shown in figures 7a and 7c, with a slope exceeding 0.97. For medium pulses, the standard deviation also shows a high correlation (figure 7b), with a slope of 0.983, demonstrating

the model's ability to capture the overall trend. Compared to 72.1% captured standard deviations with the traditional time-

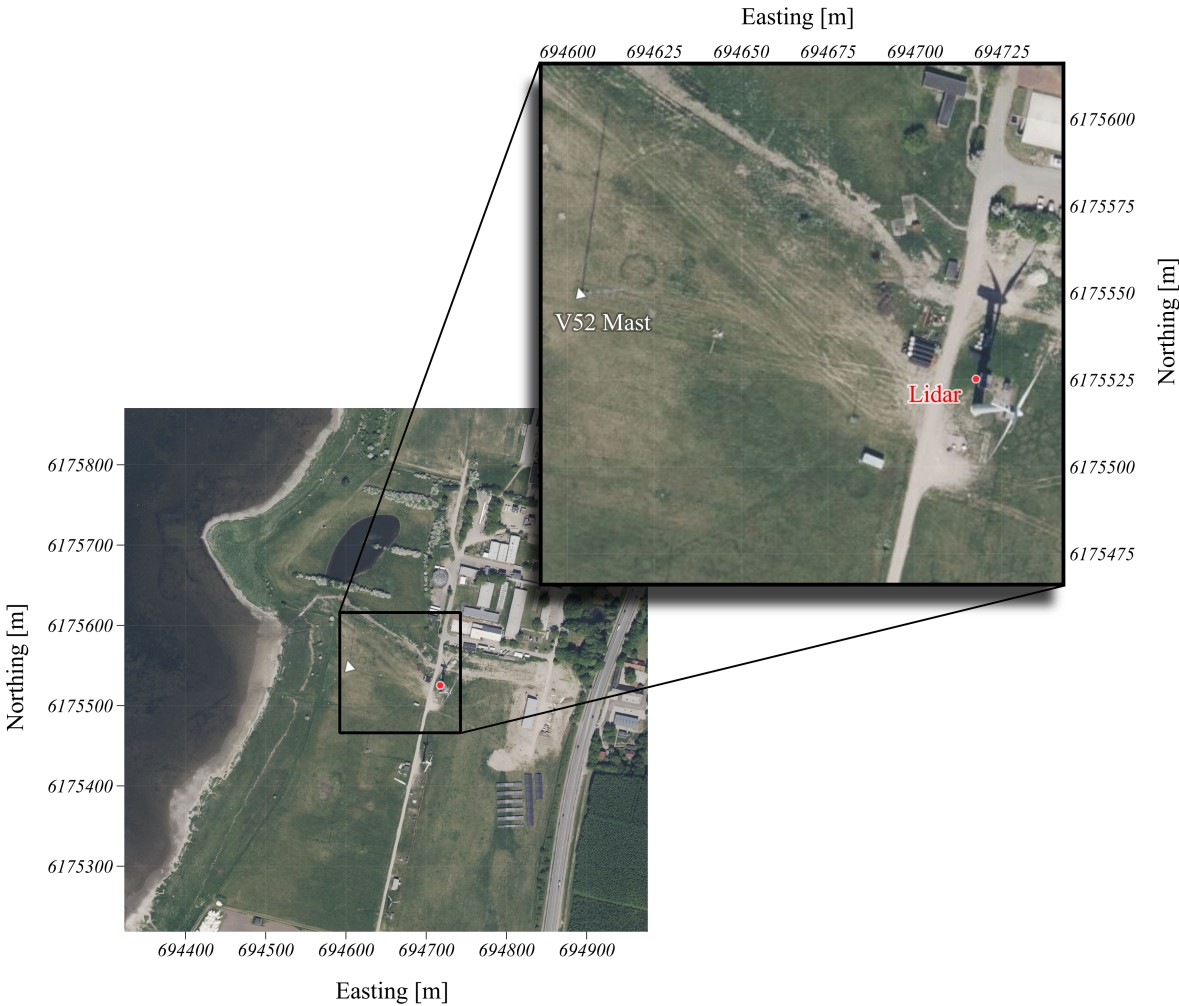

**Figure 4.** Satellite image of the experimental field at Risø campus (Denmark) provided by Bing Maps, copyright 2025 © Microsoft. The lidar is located near the V52 wind turbine (red dot), pointing toward the V52 mast (white triangle). The reference system selected in the Leica Total Station is WGS 84 / UTM Zone 32N.

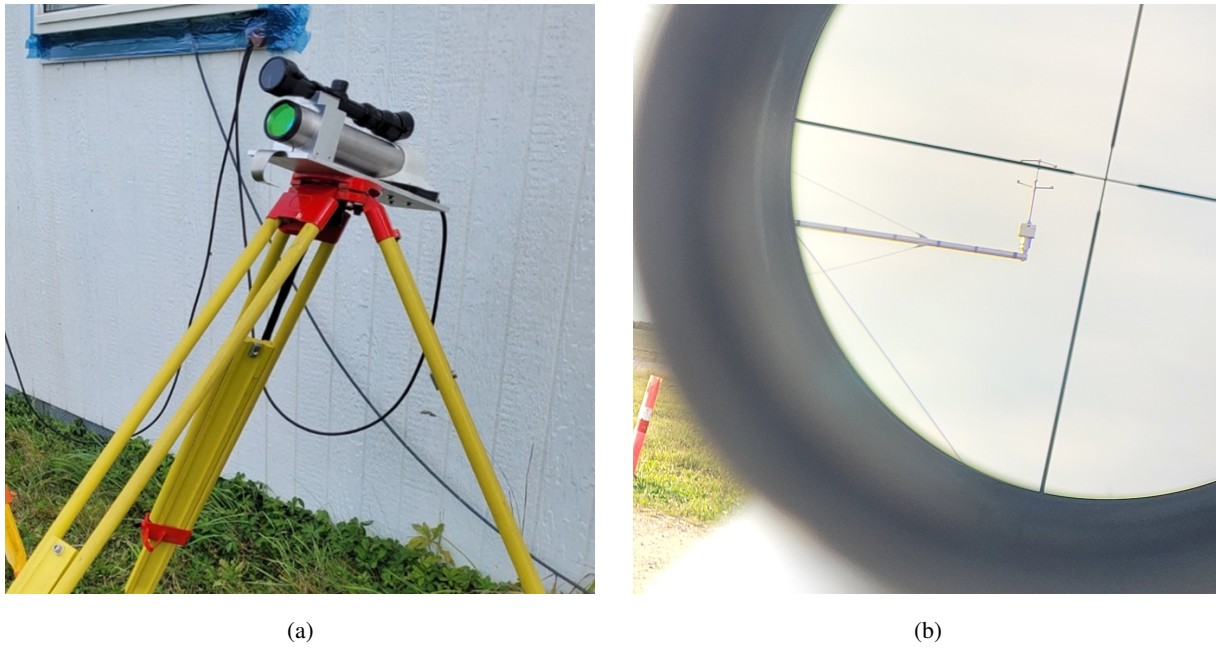

|     |     |
| :-: | :-: |
| (a) | (b) |

**Figure 5.** The prototype pulsed lidar system with a telescopic sight (a), during the process of aligning the lidar beam direction toward a sonic anemometer near the V52 mast seen through the telescopic sight (b).

domain method (figure 8), the proposed model in the frequency domain observed 98.3% of standard deviations relative to the sonic anemometer. However, the dispersion of the data indicates slight deviations from the standard deviation of the reference instrument. In cases of low standard deviations, estimations from the time domain are more accurate (figure 8), as the proposed model fit occasionally produces zero values. In contrast to medium pulses, the model's performance decreases significantly

for short pulses (figure 7d), especially for standard deviations below 1 m/s, which appear randomly distributed. In general, the model's ability to estimate wind velocity variance decreases when using short pulses, reflecting reduced effectiveness under lower frequency resolution conditions. As mentioned before, velocity variance is related to the width of the Doppler spectrum, which represents how energy is distributed across frequency bins. At low-frequency resolution, the energy distribution is not well captured: most of the spectral energy is concentrated in a single frequency bin, while the remaining bins show nearly zero

energy. In contrast, at high-frequency resolution, the energy spreads across multiple frequency bins, allowing the width of the Doppler spectrum to be estimated more accurately (from several points). As a result, when the Doppler spectrum is poorly resolved, the corresponding velocity variance estimates are expected to be less accurate.

Figure 9 demonstrates the influence of frequency resolution on the standard deviation estimate for short pulse durations (200 ns). The estimated standard deviations are presented against the normalized difference between the Doppler frequency and the

nearest frequency bin, $f_l$. A noticeable pattern emerges far away from the bin frequency, although the reason for this behavior remains unclear. Additionally, when the Doppler frequencies become close to the midpoint between two consecutive frequency

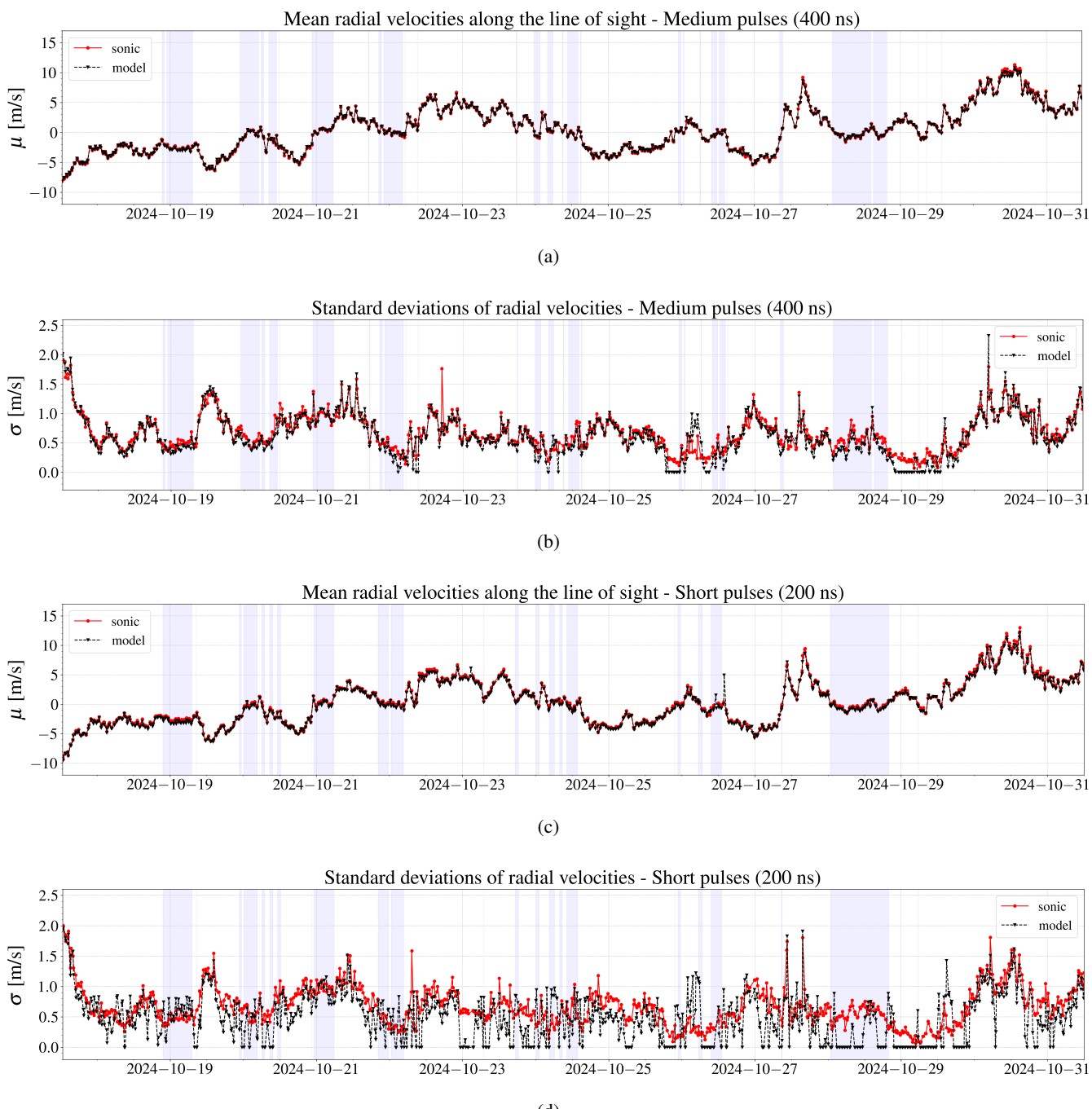

**Figure 6.** The time series for the 10-minute average radial wind velocities (a) and standard deviations (b), obtained from the model (black triangles), are based on power spectral density from the instrument's 400 ns pulses. Similar plots for the 10-minute average radial wind velocities (c) and standard deviations (d) are derived using 200 ns pulses. Each of these quantities is compared to the corresponding values from the projected velocity of the sonic anemometer along the lidar beam direction (red circles). The wake sector is shown as a shaded area.

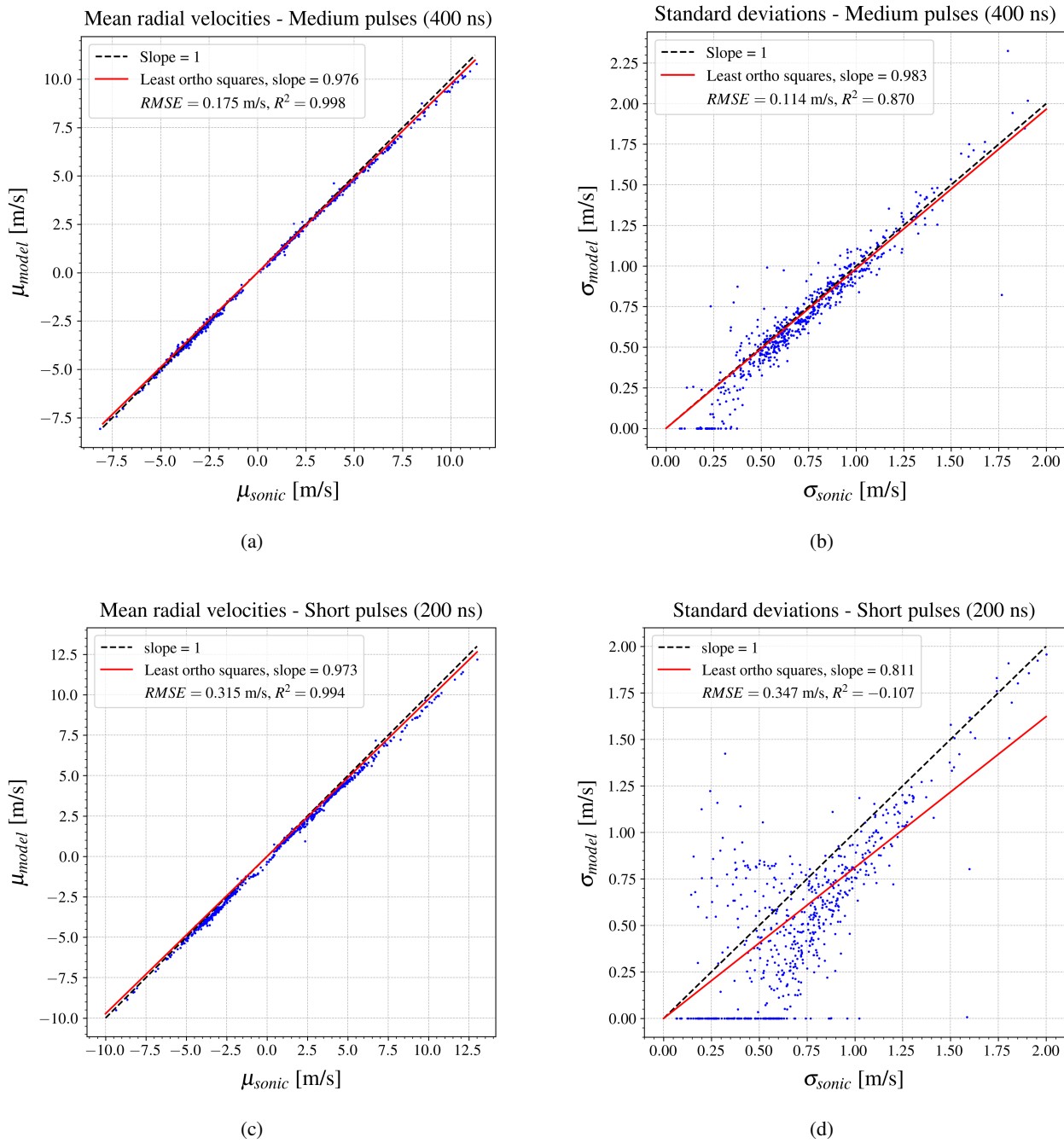

**Figure 7.** The least orthogonal squares fit between the sonic anemometer and the model, constrained to pass through zero, is shown for mean radial wind velocities (a) and standard deviation (b) using 400 ns pulses, as well as for mean radial velocities (c) and standard deviation (d) using 200 ns pulse lengths. The wake sector is excluded from regression plots.

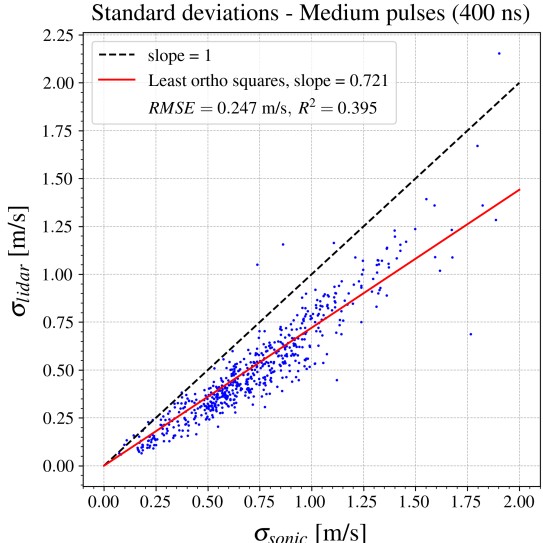

**Figure 8.** Standard deviations estimated from radial velocity time series obtained from medium pulse length (400 ns). Doppler spectra are averaged over 16,000 pulses, and a Gaussian function is fitted to each spectrum to estimate the high-frequency radial velocity time series. The standard deviations are then calculated from the velocity time series over 10-minute intervals, following the conventional approach. The wake sector is disregarded for comparison with the sonic anemometer.

bins, where the spectral gap exists, the model tends to overestimate the standard deviations compared to the sonic reference measurements.

As previously mentioned, the proposed model assumes Gaussian pulse shapes, although the actual measured pulses exhibit
deviations and skewness from Gaussian fits. To assess the impact of uncertainty in the pulse shape, variations in pulse widths are analyzed. Figure 10 compares the mean differences between the velocity standard deviations obtained from the model and sonic measurements when the assumed model pulse widths range between 149 ns and 169 ns. The results indicate that the mean difference of the standard deviations changes almost linearly at a rate of $0.003 \text{ ms}^{-1}/\text{ns}$ , which is small enough to be considered negligible. In the model, the medium pulse width was set at 159 ns based on a Gaussian fit to the temporal pulse
shape. When the pulse width is smaller than 159 ns, the estimated standard deviation error is higher compared to the reference sonic instrument. As the pulse width increases, the error decreases until reaching 171 ns, after which it begins to rise again. However, the higher error near 159 ns compared to the 171 ns pulse width may be attributed to the influence of other parameters or the simplifying assumptions used in the model. A similar analysis is conducted for the mean differences between the average radial velocities from the model and the sonic measurements, but the observed rate of change remains close to zero within this
range of pulse lengths.

Another potential limitation of this approach could be reduced performance under conditions of low signal-to-noise ratio (SNR). In this experiment, we are analyzing the short range of 120 m, where the signal strength consistently exceeds the noise standard deviation by more than two orders of magnitude, leading to a high SNR with a distinct spectral peak. While

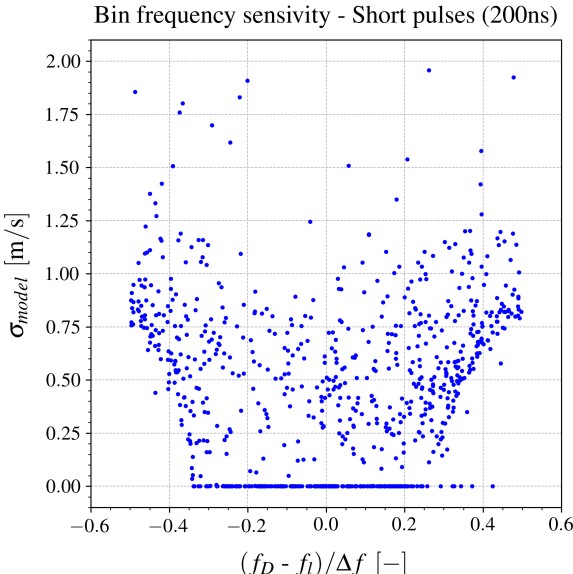

**Figure 9.** The model's estimated standard deviations as a function of the normalized difference between the Doppler frequency and the nearest frequency bin, $f_l$, for short pulse durations (200 ns).

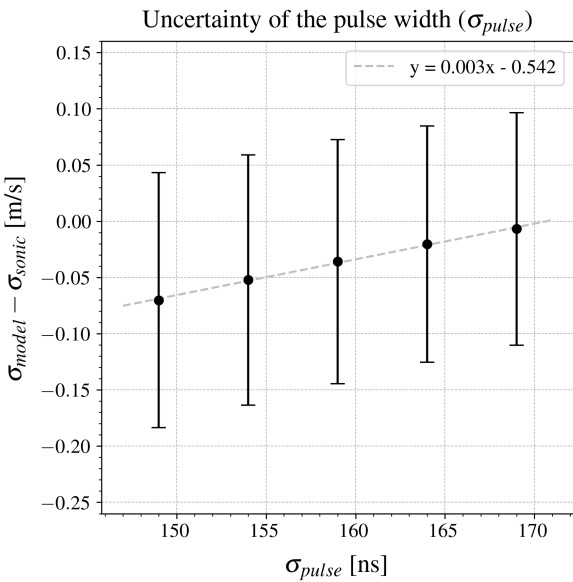

**Figure 10.** The effect of the pulse length uncertainty on the estimated standard deviations from the model. A pulse width of 159 ns, derived from a Gaussian fit to the measured temporal pulse, is used for modeling. Black circles in this plot, represent the mean differences between velocity standard deviations from the model and sonic measurements. Error bars indicate the standard deviations of the differences in the velocity standard deviation time series.

our measurements also include Doppler spectra with low SNR from longer ranges, the sonic anemometer, used as a reference instrument, is not available for these longer distances. As a result, the model's performance dependence on SNR could not be assessed directly. Therefore, we introduced artificial noise into the normalized averaged Doppler spectra to evaluate the impact of signal strength. This noise is sampled from frequencies below 40 MHz in the Doppler spectra, ensuring that it remains outside the Doppler frequency range, which is centered around the intermediate frequency of 79.72 MHz. The noise spectral density is found to follow a Weibull distribution when sampled from normalized averaged Doppler spectra. The noise vector is then randomly selected from this Weibull distribution. The normalized averaged spectra are scaled by factors ranging from 1 to 256. To analyze the impact of SNR on model performance, a constant random noise is added to all scaled spectra. The model is then fitted to the various scaled normalized spectra.

The mean differences between the time series obtained from the model and sonic measurements for the mean and standard deviation, respectively are presented in figure 11a and figure 11b for different SNR levels. SNR is here defined as the ratio of peak power density to the noise standard deviation. The first sample in each plot represents the differences with the original model estimations without additional noise. The results show that the differences between estimated and measured mean radial wind velocities for different SNR levels remain relatively constant until the SNR drops significantly. The deviation in the mean difference begins when the signal is 64 times weaker (SNR = 14.6) and significantly increases after that. However, the differences in standard deviations between the model and measurements are more sensitive to noise levels, with deviations from the original results starting at an SNR of 117.1. This outcome is expected, since noise affects the Doppler spectrum width more significantly than the mean frequency at low SNR levels.

Another anticipated source of uncertainty is the velocity gradient along the beam, which was not included in the model to maintain simplicity. In this experiment, when the velocity gradient was assessed across two neighboring range gates using lidar measurements, it showed a low gradient along the beam, resulting in no significant impact on the accuracy of the variances estimated. However, in the presence of strong wind shear, the effect of the velocity gradient along the beam could become relevant to account for in the modeling.

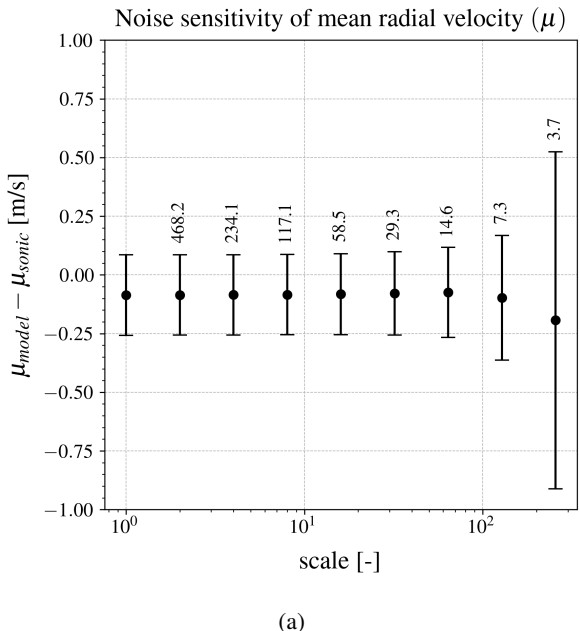

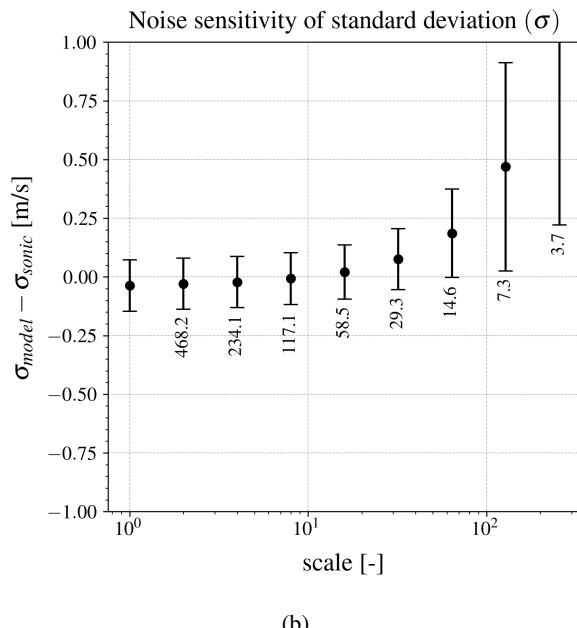

|  |  |
|---|---|
| (a) | (b) |

**Figure 11.** Sensitivity of the estimated mean (a) and standard deviation (b) to SNR. The normalized averaged spectra were scaled by factors ranging from 1 to 256, after which a constant random noise was added to all scaled normalized spectra to examine the impact of SNR. The mean and standard deviation of the differences in statistics are represented by circles and error bars, respectively, with SNR values indicated around the tails.

## 5   Conclusion

This study introduced a novel approach for estimating turbulence statistics from a pulsed Doppler lidar, aiming to reduce the effects of probe volume averaging and high-frequency noise in the velocity estimation process. The proposed method derives the mean and standard deviation of the radial velocities directly from the frequency domain, using averaged Doppler spectra. This approach is developed on the basis of a model for the ensemble-averaged Doppler spectra from beaten Gaussian pulses within the range gate. To validate the model, a field experiment was conducted at the DTU Risø campus in Denmark using a ground-based pulsed lidar pointing toward a sonic anemometer mounted at 18 m height and at a 117.45 m radial distance. Doppler spectra were collected from October 17 to October 31, 2024, alternating nominal pulse lengths between 200 ns (short) and 400 ns (medium) every 20 minutes. The model was fitted to 10-minute averaged spectra to estimate radial mean velocities and standard deviations.

Following comparison with the sonic anemometer, the model showed strong agreement with the reference instrument in estimating the standard deviations using medium pulses, as well as the mean radial velocities using both short and medium pulses. The estimated mean radial velocities demonstrate the least orthogonal square regression slopes exceeding 0.97 for short and medium pulses. The estimated standard deviations using medium pulses provide a slope of 0.983, indicating satisfactory performance. While the traditional time-domain approach captures 72.1% of the standard deviations, the proposed model achieves 98.3% of the standard deviations compared to the reference instrument. However, when the actual standard deviation is low, the model fit sometimes estimates it as zero; in such cases, the time-domain estimate proves to be more accurate. In addition, this model performs poorly for Doppler spectra from short pulses, highlighting the critical role of frequency resolution for accurate variance retrieval. The sensitivity of the LOS standard deviation estimates to the uncertainty in the pulse width within $\pm 10$ ns was also assessed, with errors varying at a negligible rate of 0.003 ms$^{-1}$/ns. Furthermore, the robustness of the method under low SNRs was evaluated by adding artificial noise to the Doppler spectra. The results indicate a deviation in performance starts when the signal is weakened by factors of 64 for mean and 8 for standard deviation estimation.

Although the current study offers valuable information on measuring the standard deviation of radial velocities using a medium pulse of a Doppler lidar, future research should investigate potential performance enhancements, particularly through the application of shorter pulses. Further refinement may involve reformulating the current methodology using a skewed distribution that more accurately represents the actual measured pulse shape in the nominal 200 ns pulse duration. Despite some limitations for this study, the proposed model provides reliable LOS turbulence variance estimates from a pulsed lidar using a 400 ns nominal pulse length, offering a more precise turbulence characterization, which might be useful for wind energy applications.

*Data availability.* Data will be provided upon request, subject to the decision of the Technical University of Denmark (DTU) and Lumibird.

*Author contributions.* MM: draft, methodology, software, and analysis. JM: conceptualization, methodology, software, and funding acquisition. MS: supervision, analysis, editing, review, feedback, and funding acquisition. GL and GG: supervision, review, and funding acquisition.

*Competing interests.* MM, GL, and GG are employed by Lumibird SA.

255  *Acknowledgements.* This project has received funding from the European Union's Horizon Europe research and innovation program under the Marie Skłodowska-Curie grant agreement No 101119550. The authors acknowledge Gunhild Rolighed Thorsen, Per Hansen, Karen Enevoldsen, Michael Courtney, Ida Egholm Nielsen, Steen Arne Sørensen, Michael Rasmussen, and other contributors from the Measurement Systems and Methods (MEM) section of DTU Wind and Energy Systems for performing field experiments and collecting the required datasets.

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
