# Peer review of "Squeezing Turbulence Statistics out of a Pulsed Doppler Lidar"

_EGUsphere, 2025_

## Author Response (AR1)

**Author Comments:**

We would like to thank the reviewers for their constructive feedback. We have carefully considered each comment and revised the manuscript accordingly. Our point-by-point responses are provided below.

**RC1: https://doi.org/10.5194/egusphere-2025-2226-RC1, 23 Jun 2025**

**Major comment:**

The article mentions in the Introduction and Conclusion sections that this method could help better characterize turbulence, but little to no detail is given about this point. I understand that this is not the main focus of the paper, but I think it could significantly benefit from more details in this respect. This would allow the reader to get an idea of the particular physical phenomena that the measurements of this method would be targeting.

Some questions associated with this point are: What is the type of turbulence that this method would allow to study? Is it isotropic turbulence? What spatial and temporal scales would benefit from these measurements? Would these scales correspond to the inertial subrange? Viscous subrange?

Answering these questions would add value to the paper, as they add context, not only encapsulating it to be "useful for wind energy applications".

*Answer:*

Thank you for your comment. In response to the RC, the following sentences have been added to the manuscript:

"The proposed model allows for the quantification of turbulence statistics across all scales. A significant advantage of this approach is its ability to capture velocity variances at scales smaller than the probe volume, a feature generally absent in the conventional method. The model considered the homogeneity assumption along the LOS, which neglects the shear effects. However, it does not rely on any isotropy assumptions that restrict the measurements to preferred directions."

**Minor comments:**

Line 50: What attribute of the pulses has a Gaussian temporal profile? Please clarify.

*Answer:*

The laser power has approximately a Gaussian temporal profile (as illustrated in Figure 2). To address the RC, the Figure's caption is modified as follows:
"Measured laser power over time (blue points) with Gaussian fits (red curves) approximating the pulse shapes. Each pulse is normalized to its peak power."

Line 55: Please expand on or provide a citation for the term "interrogation window." As far as I'm aware, this is not widely used terminology.

*Answer:*

Thanks to the reviewers for recommending the use of standard terminology. Accordingly, we replaced "interrogation window" with the standard term "range gate" in the manuscript.
* * *
Line 140: "However, when the standard deviation is low, the model occasionally estimates zero." Is this an expected behavior? What causes it? This seems to be the main source of error in your comparison.

*Answer:*

Thank you for your comments. While we anticipate greater challenges in estimating velocity variances under laminar flow conditions, observing zero variances is not expected. To clarify this point and explain other sources of errors, we have added the following sentences to the manuscript:

"In particular, under laminar flow conditions, estimating velocity variances is expected to be challenging, since in this regime the primary contribution to the Doppler spectral width arises from the finite length of the range gate. However, obtaining zero variances from the model is clearly unrealistic. This outcome reflects the imperfection of the model, which relies on several simplifying assumptions. One key assumption is that the pulse shape follows a Gaussian distribution, whereas, in practice, measured pulses often deviate from a Gaussian form. Another source of error arises from the background noise removal: when the noise level is subtracted, it can sometimes yield negative values in the Doppler spectra. The negative Doppler spectra are non-physical and introduce inaccuracies in the variance estimation."
* * *
Line 150: Why does the model show reduced effectiveness under conditions of lower frequency resolution?

*Answer:*

To clarify the influence of frequency resolution, the following sentences are added to the manuscript: "As mentioned before, velocity variance is related to the width of the Doppler spectrum, which represents how energy is distributed across frequency bins. At low-frequency resolution, the energy distribution is not well captured: most of the spectral energy is concentrated in a single frequency bin, while the remaining bins show nearly zero energy. In contrast, at high-frequency resolution, the energy spreads across multiple frequency bins, allowing the width of the Doppler spectrum to be estimated more accurately (from several points). As a result, when the Doppler spectrum is poorly resolved, the corresponding velocity variance estimates are expected to be less accurate."

The paper is a useful advance in a field of current relevance. The results are generally clearly presented, but the description of the methodology confused me in places. I have some specific suggestions below of areas for possible improvement. Also, some of the experiments details should be clarified. The paper would be enhanced with a small amount of additional information and analysis.

Specific comments and suggestions below:

On P4, the parameter "s" needs to be more clearly defined. Eqns 1 and 3 imply it has dimensions of time, but line 65 states it is a position and furthermore, line 67 states it "has a concise velocity", which confused me further. Possibly s is the range (distance) to the scatterer and the corresponding time in the eqns should be 2s/c? However, Figure 3 introduces another parameter x defined by t=x/c, but x does not appear elsewhere in the document.

*Answer:*

Thanks to the reviewers for pointing out this inconsistency. Parameter "s" is related to the position of the particle; Therefore, line 65, Equations 1, 3, and 7, as well as Figure 3, are modified accordingly.

What is the height of the lidar relative to the sonic? If not the same, then would horizontal beam path possibly give better results?

*Answer:*

Thank you for this question. In this experiment, the pulsed lidar system was positioned on the ground and directed toward a sonic located at 18 meters, making the relative height approximately 18 meters. For the comparison with the sonic measurements, the three-dimensional wind vector was projected onto the lidar beam direction. Under the assumption of a perfect projection, no preferred direction should be expected in this method.

What is the likely sensitivity to wind direction: we could expect better agreement if the wind aligns along the beam? P6 line 119 implies the measurements were taken with "westerly inflow": were the data filtered for a range of wind directions around West. If not, what was the distribution of wind directions during the measurement period, and could the analysis be performed with different levels of direction filtering to examine the impact?

*Answer:*

Building on the previous question, no preferred direction should be expected in this method. To illustrate this, the regression plots are colored by wind direction (shown below in Figures a and d), with further analysis presented using wind rose plots. The rose plots (Figures b and e) show the frequency of mean wind direction from sonic anemometers versus the distribution of standard deviations from short and medium pulses, respectively. Bins with very low variances (shown in black, representing zero estimates from the model) are distributed across different wind directions with no specific patterns. In addition, the distribution of the difference between the sonic and model-derived standard deviations is shown as a function of wind direction (Figures c and f). The plot for the short pulse indicates that larger errors

(orange, red, dark red, and black) occur across various wind directions. A similar result is also observed for the medium pulse.

P6 line 119 states the westerly flow, which refers to the optimal wind direction with respect to the geometry of the experiment (shown in Figure 4). The westerly flow comes from the fjord, and it is not distorted by the V52 wind turbine and the met mast. To address RC, the following sentences are included in the manuscript:

"The optimal wind direction according to the geometry of the experiment is the westerly sector, where the flow comes. The westerly flow comes from the fjord and remains unaffected by wake effects from either the V52 wind turbine or the met mast."

Following further analysis, a filter was applied to the 170°–200° and 90°–110° sectors to ensure the validity of the comparison with the sonic anemometer by removing wake sectors. These sectors are shown as a shaded area in the time series (Figure 6) and subsequently excluded from the regression plots (Figure 7). The following sentences are added in this regard:

"Since measurement errors may occur under easterly and southerly winds due to wake effects from the wind turbine and the met mast, data from the 90°–110° and 170°–200° sectors were excluded to ensure the validity of the comparison between the lidar and the sonic anemometer."

[Figure]

Assumption of Gaussian turbulence spectrum (Eqn 5, p9 line 163): can we see some example spectra? Or possibly some analysis of higher order statistics to indicate the validity of the assumption?

*Answer:*

Thank you for your comment. Equation 5 assumes that the wind velocity follows a Gaussian distribution, which is commonly used to describe the probability distribution of stochastic variables. This assumption has been widely adopted in turbulence models for wind energy (Mann, 1994). We acknowledge that the wind velocity distribution is not strictly Gaussian and exhibits deviations from Gaussianity. Nevertheless, the purpose of this study is not to evaluate whether turbulence is truly Gaussian; rather, the assumption is used to derive a model for average Doppler spectra and to estimate one-point statistics.

On page 9, line 163, "As previously mentioned, the proposed model assumes Gaussian pulse shapes, although the actual measured pulses exhibit deviations and skewness from Gaussian fits.", we address the Gaussian pulse shape illustrated in Figure 2. This should not be mistaken for the velocity probability distribution given in Equation 5.

*Mann, J. (1994). The spatial structure of neutral atmospheric surface-layer turbulence. Journal of Fluid Mechanics, 273, 141–168. doi:10.1017/S0022112094001886*
* * *
P8: what causes the zero values of stdvn? The authors should attempt at least a tentative explanation, e.g. negative (unphysical) values brought about by noise, rounded up to zero?

*Answer:*

Thank you for this comment. The following explanations are included to the manuscript: "In particular, under laminar flow conditions, estimating velocity variances is expected to be challenging, since in this regime the primary contribution to the Doppler spectral width arises from the finite length of the range gate. However, obtaining zero variances from the model is clearly unrealistic. This outcome reflects the imperfection of the model, which relies on several simplifying assumptions. One key assumption is that the pulse shape follows a Gaussian distribution, whereas, in practice, measured pulses often deviate from a Gaussian form. Another source of error arises from the background noise removal: when the noise level is subtracted, it can sometimes yield negative values in the Doppler spectra. The negative Doppler spectra are non-physical and introduce inaccuracies in the variance estimation."
* * *
P8: I have a small objection to the phrase "captures 96.7% of stdvn" since there may be other contributions present, for example, instrumental and other noise sources that are nothing to do with the wind.

*Answer:*

Thank you for your comment. Because the Doppler spectra are averaged over 10-minute intervals, the resulting statistics are expected to be more stable, with less influence from random noise (as shown in Figure 1). Also, it's assumed that the sonic anemometer provides a reliable measurement as a reference instrument. To clarify that this metric is related to the comparison, the manuscript is modified with the following sentences:

"In comparison to the conventional time-domain approach, which accounts for only 72.1% of the standard deviation, the proposed spectral method captures 98.3% of the standard deviation observed in the sonic anemometer."

"Compared to 72.1% captured standard deviations with the traditional time-domain method (figure 8), the proposed model in the frequency domain observed 98.3% of standard deviations relative to the sonic anemometer."

"While the traditional time-domain approach captures 72.1% of the standard deviations, the proposed model achieves 98.3% of the standard deviations compared to the reference instrument."
* * *
P14, fig 11, clearly explain "scale" axis – presumably the level of added noise?

*Answer:*

Thank you for your comment. The normalized averaged spectra are scaled by factors ranging from 1 to 256, after which a constant random noise is added to all scaled spectra to study the effect of different signal-to-noise ratios. To address the RC and improve clarity, we will include this explanation in the caption of Figure 11:

"The normalized averaged spectra were scaled by factors ranging from 1 to 256, after which a constant random noise was added to all scaled normalized spectra to examine the impact of SNR."